# Bilateral Nephroblastoma with Dilated Cardiomyopathy as an Indication for Off-Protocol Treatment: A Case Report

**DOI:** 10.3390/ijerph17249483

**Published:** 2020-12-18

**Authors:** Patrycja Sosnowska-Sienkiewicz, Ewelina Gowin, Katarzyna Jończyk-Potoczna, Przemysław Mańkowski, Jan Godziński, Danuta Januszkiewicz-Lewandowska

**Affiliations:** 1Department of Pediatric Surgery, Poznan University of Medical Sciences, 60-572 Poznan, Poland; mankowskip@ump.edu.pl; 2Department of Health Promotion, Poznan University of Medical Sciences, 60-572 Poznan, Poland; ewego@ump.edu.pl; 3Department of Pediatric Radiology, Poznan University of Medical Sciences, 60-572 Poznan, Poland; jonczyk@ump.edu.pl; 4Department of Pediatric Traumatology and Emergency Medicine, Wroclaw Medical University, 50-041 Wroclaw, Poland; jgodzin@wp.pl; 5Department of Pediatric Oncology, Hematology and Transplantology, Poznan University of Medical Sciences, 60-572 Poznan, Poland; danuta.januszkiewicz@ump.edu.pl

**Keywords:** cardiomyopathy, children, cardiotoxicity, hypertension, Wilms tumor

## Abstract

Patients with a Wilms tumor are often admitted to the hospital accidentally, with an abdominal mass causing asymmetry of the abdominal wall. Hypertension accompanying a Wilms tumor occurs in about 10–27% of children, but cardiomyopathy associated with a Wilms tumor is very rarely described. This publication presents a case of a 9-month-old girl with a bilateral Wilms tumor accompanied by dilated cardiomyopathy since her initial cancer diagnosis, as well as her off-protocol treatment. The severe condition of the child forced the application of off-protocol treatment, i.e., accelerated resection of a larger tumor, which enabled the improvement of heart performance and made subsequent therapy possible. In the course of the presented treatment, a gradual normalization of cardiac ventricular function and contractility was observed. In conclusion, a massive abdominal tumor associated with abdominal compartment syndrome compromised the functioning of the cardiovascular system in the young child. Therefore, earlier removal of Wilms tumors in patients with heart failure should be considered. This may result in the improvement of cardiovascular function and the possibility of further therapy.

## 1. Introduction

Wilms tumors (nephroblastoma) are the most common kidney cancer in children [1]. More often, the tumor occurs in children with hemihypertrophy or children with another malformation of the genitourinary, cardiovascular, or musculoskeletal system [1]. Usually, the disease affects one kidney, but in 5–10% of cases, it is bilateral [2,3].

Wilms tumors are genetically heterogeneous and mutations in *WT1*, WTX, and *CTNNB1* underlie the genetic basis of about one-third of Wilms tumors. Another gene mutation found in Wilms tumors is the *AMER1* gene. Changes on the short arm of chromosome 11 where IGF2 and H19 genes are located were also associated with development of a Wilms tumor.

Patients with Wilms tumors are often admitted to the hospital accidentally with an abdominal mass causing asymmetry of the abdominal wall. Other symptoms present at the time of diagnosis of a Wilms tumor, such as malaise, loss of appetite, abdominal pain or hematuria, occur in 20–30% of patients [1]. Hypertension accompanying a Wilms tumor occurs in about 10–27% of children [4,5]. Cardiomyopathy associated with a Wilms tumor is very rarely described [6,7,8].

Chemotherapy and surgery, and in some cases also radiation therapy, are an effective treatment [1]. For bilateral tumors, nephron-sparing surgery, preferably bilateral, or otherwise performed at least on the less involved side, is the current standard [9].

This publication presents the case of a girl with bilateral Wilms tumors accompanied by dilated cardiomyopathy since her initial cancer diagnosis, as well as her off-protocol treatment.

## 2. Case Report

A 9-month-old girl was admitted to the Pediatric Oncology Department due to the abdominal mass noticed by her parents two weeks earlier. She had been born from a second pregnancy at 39 weeks of gestation, with birth weight of 3790 g and the APGAR score of 10. The pregnancy had gone without any complications. Her parents and older brother are healthy.

A large mass crossing the midline of the abdomen was palpable at admission. The child presented low body weight (8600 g), lack of appetite, increased respiratory and heart rate, and increased blood pressure. Computer tomography (CT) of the abdomen showed bilateral kidney tumors. On the right side, a heterogeneous pathological mass of 11.2 × 10.4 × 15.8 cm (979.1 cm^3^; acc. to calculation formula V = D cm × S cm × W cm × 0.532) was extending from the diaphragm to the pelvis. The tumor shifted the liver to the left side. In the middle part of the left kidney, three solid pathological lesions of a similar nature were visible—the first one located in the upper-middle pole, 3.0 × 1.8 × 3.2 cm (9.2 cm^3^), the second one located laterally from the first, 2.2 × 1.6 × 1.9 cm (3.6 cm^3^), and the third one slightly lower and on the side, measuring 0.6 × 1.0 × 1.2 cm (0.4 cm^3^) (Figure 1).

Lung CT did not show metastases. Echocardiography presented dilatation of cardiac ventricles with reduced ventricular function and impaired cardiac contractility—left ventricle (LV) transverse shortening fraction (FS%) was 10–12% (normal range 28–44%; mean 36%), and LV ejection fraction (EF%) was 38–40% (normal range 64–83%; average 74%). Coronary artery abnormalities were excluded. Electrocardiography (ECG) at rest showed normal sinus rhythm. The heart rate at rest was 170–180 beats per minute. Twenty-four-hour Holter monitoring showed no abnormalities. The girl has rated at 7–8 points (Ross Heart Failure Classification), which indicates moderately severe heart failure.

Chemotherapy with vincristine and actinomycin was prescribed (VCR 1.5 mg/m^2^, Actinomycin 0.045 mg/kg, all drug doses due to body weight below 12 kg according to SIOP scheme were reduced to 67%). Despite chemotherapy and cardiac treatment including spironolactone, captopril and carvedilol, the child still presented tachycardia, tachypnea and moderate hypertension with increasing levels of B-type natriuretic peptide (BNP), lactate dehydrogenase (LDH), high-sensitivity troponin T (hscTnT) and plasma renin activity (PRA).

Deterioration of heart contractions was also observed. An increase in left ventricular dilatation with reduced contractility and foramen ovale shunting was found in the following echocardiography. CT examination performed after six weeks of treatment did not show significant tumor reduction (Figure 2).

To reduce the intra-abdominal contents and counteract the abdominal compartment syndrome (ACS) associated with a large Wilms tumor, it was decided to start the surgery from the right side. Uneventful nephrectomy was performed, followed by a prolonged stay at the intensive care unit due to the cardiologic problems. Histopathological examination revealed a stromal nephroblastoma with a medium degree of differentiation. Tumor regression was low, less than 15%. The tumor was mostly stromal, with spindle-like rhabdomyomatous and smooth muscle cells. The kidney tissue was dysplastic, partly cystic.

A CT scan one week after surgery showed persistent tumors in the left kidney, the first one of 2.7 × 1.8 × 2.4 cm (6.2 cm^3^), the second one of 0.9 × 0.7 × 1.6 cm (0.5 cm^3^) and the third one of 0.7 × 0.9 × 0.5 cm (0.2 cm^3^). Due to cardiomyopathy, doxorubicin was omitted, and chemotherapy with etoposide and carboplatin was continued (etoposide 150 mg/m^2^ and carboplatin 200 mg/m^2^, all drug doses due to the body weight being below 12 kg according to the SIOP scheme were reduced to 67%). At week 12 of chemotherapy, magnetic resonance showed that the three lesions in the left kidney were still present. Nephron-sparing surgery was performed. A wedge resection of all three lesions was performed. The postoperative course was uneventful. All three lesions were removed. The dominating one appeared to be a stromal nephroblastoma with spindle cells and rhabdomyomatous component. No necrotic features were found. Two smaller ones were the foci of kidney dysplasia.

In the course of the presented treatment, a gradual normalization of cardiac ventricular function and contractility was observed. The EF% values at the time of diagnosis and further in the course of preoperative chemotherapy were 38–40%. After the first surgical procedure and further in the course of postoperative chemotherapy, EF% rose to 50–55%. The last EF% after the completion of the entire postoperative chemotherapy was 60–65%. Changes in the levels of hscTnT, BNP, PRA and LDH during the hospitalization are illustrated in Figure 3.

## 3. Discussion

Hypertension and cardiomyopathy may coexist with a Wilms tumor [6]. Hypertension may be caused by the mechanical compression of the renal artery or renin production by the cancer cells. The pathophysiology of renovascular hypertension is due to an initial reduction in renal perfusion that occurs as a result of stenosis of the main renal artery or one of its branches. Heart failure is explained as secondary to the secretion of renin and angiotensin II, a vasoactive agent other than renin or catecholamines [10,11,12]. Hence, Trebo et al. recommend routine echocardiography for patients with a suspected Wilms tumor [6]. Not only does this allow the detection of heart disease associated with a Wilms tumor, but it also enables monitoring of the toxicity of chemotherapy.

A 9-month-old patient treated in our center was admitted to the hospital with bilateral Wilms tumors with hypertension and concomitant cardiomyopathy already present at the beginning of hospitalization. Despite chemotherapy and cardiology treatment, deterioration of heart function was observed. Both the large tumor itself and the chemotherapy used can be responsible for heart failure. It was decided on the necessity of prompt surgical treatment. After removal of the right kidney with a massive tumor, the parameters of cardiac function gradually improved despite the ongoing chemotherapy. As Stine et al. described in their publication, cases of heart failure and bilateral Wilms tumor are a severe problem in treatment. They recommend surgical treatment of unilateral tumors [4]. We did the same for our girl, which was beneficial. 

Several studies have emphasized the important role of surgery in improving the hemodynamic function of the heart. Chavalon et al. described a case of a 7-month-old girl with a unilateral Wilms tumor. After chemotherapy and tumor removal, heart function improved [7]. Sethasathien et al. also reported that cardiac activity normalized after Wilms tumor resection despite chemotherapy and radiation therapy [13]. Parry et al. presented a case of a child with cardiomyopathy and unilateral Wilms tumor without hypertension, but with an elevated level of renin and angiotensin. In this case, the chemotherapy and surgery used allowed cardiac function to return to normal [8].

Miura et al. described a case of a 3-month-old girl with severe hypertensive heart failure and Wilms tumor with hyperreninemia. The child received mechanical ventilation and antihypertensive treatment. The authors emphasized the importance of preoperative hemodynamic control and sedation that allow safe surgical treatment [14]. Our patient did not require sedation before surgery; her condition was stable. Similarly, Ganigara et al. presented a case report of neuroblastoma accompanied by dilated cardiomyopathy without elevated levels of catecholamines. The reason for hypertension was probably the compression of the renal artery by the huge tumor, which resulted in high levels of serum renin and angiotensin [12].

The treatment of bilateral renal tumors is frequently challenging. General policy emphasizes maximum effort to perform nephron-sparing surgery and complete resections on both sides [1,9]. Unilateral partial nephrectomy and contralateral total nephrectomy may also be the right solution if sufficiently functional renal tissue is preserved. In sporadic, dramatic cases, complete bilateral nephrectomy followed by dialysis and kidney transplantation is an ultimate solution. It is generally agreed that patients with bilateral nephroblastoma should be operated on after neoadjuvant chemotherapy. Its duration, however, should not be too long. Preoperative chemotherapy longer than 12 weeks offers further benefit only exceptionally. Lack of response to chemotherapy may suggest both high-risk (anaplastic) and intermediate-risk (stromal predominant) histological subtypes of nephroblastoma [15]. Classical subtypes have usually responded well. However, some of the good responders do not demonstrate any marked decrease in size, but the tumor turns necrotic. Regarding the order of intervention, the general rule has suggested that the more accessible side would be the one to be operated on first. In the described case, severe cardiomyopathy was a result of the larger mass on the right, less accessible side, and posed an immediate vital risk. That is why the life-saving resection of the bigger tumor in the presented case appeared appropriate.

The cited publications were useful in making treatment decisions for our patients [16,17]. In most cases, the publications describe patients with unilateral Wilms tumors. In our case, we described an infant with bilateral Wilms tumors with dilated cardiomyopathy and hypertension associated with hyperreninemia.

## 4. Conclusions

In conclusion, a massive abdominal tumor associated with the abdominal compartment syndrome negatively affected the cardiovascular system, especially in such a young child. Therefore, earlier removal of Wilms tumors in patients with heart failure should be considered. This may result in the improvement of cardiovascular function and the possibility of further therapy.

## Figures and Tables

**Figure 1 ijerph-17-09483-f001:**
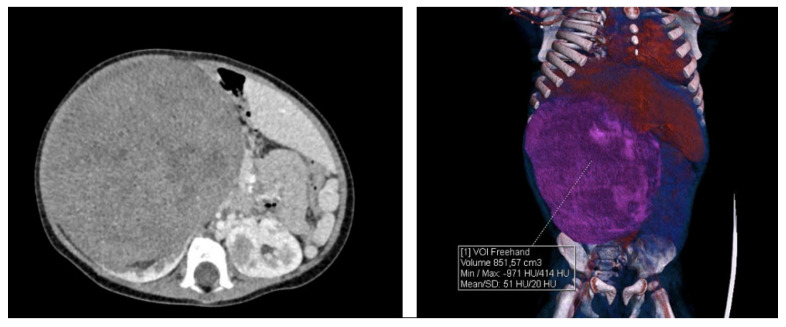
CT scan of the abdomen on admission to the hospital.

**Figure 2 ijerph-17-09483-f002:**
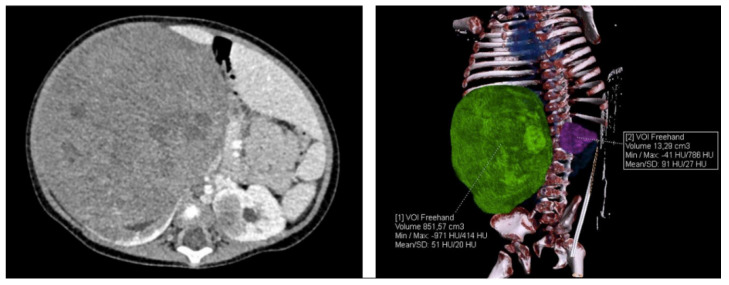
CT scan of the abdomen after 6 weeks of preoperative chemotherapy.

**Figure 3 ijerph-17-09483-f003:**
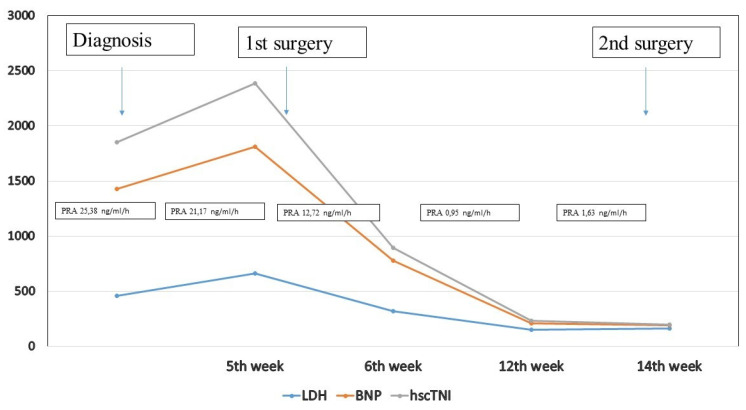
Levels of hscTnI, BNP, PRA and LDH in the course of treatment. Range of normal levels: LDH 120–230 IU/L, BNP 0.5–30 pg/mL, hscTnI <5 ng/L, PRA 0.3–1.90 ng/mL/h.

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
