# Peer review of "Bilateral Nephroblastoma with Dilated Cardiomyopathy as an Indication for Off-Protocol Treatment: A Case Report"

_ijerph, 2020, doi:10.3390/ijerph17249483_

Round 1

Reviewer 1 Report

It is a well written case report about the therapeutic experience of Wilms tumor and its related heart effects.

After the surgeries and chemotherapy, the data of EF changes needs to be presented. It is mentioned in the context, “In the course of presented treatment, a gradual normalization of cardiac ventricular function and contractility was observed.” (line 93-94). Please present the changes of EF during and after the course of therapy. When did the EF return to normal?

Levels of hscTnI, BNP, PRA and LDH decreased after the first surgery. Are there any changes after the second surgery? Did they remain low?

Author Response

Dear Reviewer,

We apologize for the delay in responding. The only explanation for this is the multitude of duties in our work in the hospital and with medical students.

Thank you for your time to prepare a review of our article. Please find our answers and comments:

  1. The manuscript has been sent for linguistic correction, and we attach the manuscript with the corrections made and the version already corrected.
  2. hscTnl, BNP, PRA and LDH levels decreased dramatically after the first surgery and further in the course of postoperative chemotherapy (Fig.3). The girl was operated on for the second time with hscTnl, BNP, PRA and LDH levels already within normal limits. These parameters have remained within normal limits until today.
  3. The EF% values at the time of diagnosis and further in the course of preoperative chemotherapy were 38-40%. After the first surgical procedure and further in the course of postoperative chemotherapy, EF% rose to 50-55%. The last EF% after the completion of the entire postoperative chemotherapy was 60-65%. A paragraph on EF% changes has been added to the manuscript.

Regards,

Danuta Januszkiewicz-Lewandowska, prof. MD

Patrycja Sosnowska, MD

Reviewer 2 Report

  1. The paper needs extensive editing of English. There were major grammatical errors, inappropriate word choices, and unusual sentence structure throughout the paper.
  2. What is the prevalence of cardiomyopathy in Wilms tumor?
  3. Please provide a summary of previous publications that described dilated cardiomyopathy in the context of Wilms tumor. How is your case report different from previous case reports? What new information does your case report add to the current knowledge of Wilms tumor?
  4. Please add a paragraph in the introduction about the genetic basis of Wilms tumor.
  5. Did the authors look into this patient's family member (outside of their immediate family) for the existence of a relative with Wilms tumor? This could suggest a genetic predisposition and perhaps further workup may be beneficial.
  6. Was there an echo done post-op? The authors mentioned: "In the course of presented treatment, a gradual normalization of cardiac ventricular function and contractility was observed.", please provide objective echo parameters to support this statement. 
  7. Was the patient hypertensive at presentation? What was their BPs at presentation and throughout the course of treatment?

Author Response

Dear Reviewer,

We apologize for the delay in responding to the Reviewer's comments. The only explanation for this is the multitude of duties in our work in the hospital and with medical students.

Thank you for Your time to prepare a review of our article and your insightful but valuable comments.

  1. The manuscript has been sent for linguistic correction, and we attach the manuscript with the corrections made and the version already corrected.
  2. The prevalence of Wilms tumor is 1/10000 children aged 0-15 years. When analyzing the literature review of PUBMED from the last 30 years only 8 case reports were found. To summarise, cardiomyopathy seems to be very rare, so it is difficult to estimate the prevalence of cardiomyopathy at the time of diagnosis of nephroblastoma.
  3. The authors quote extensively in the Discussion the literature data describing the occurrence of cardiomyopathy in the Wilms tumor. Therefore, we believe that it is difficult to add another paragraph in the Introduction. Nevertheless, in 2020 two case reports were presented (we added these papers in the bibliography; 16, 17), which summarize in the table exactly the same data of 7 patients with cardiomyopathy and Wilms tumor.
  4. So far, only one case with a bilateral Wilms tumor has been described in the literature (Stine et al., 1986). In the remaining case reports, renin increase was not always observed. Our case is presented in order to emphasize the necessity of preliminary ECG, ECHO examinations in every child with Wilms tumor. However, the most important message of our observation is that the decision of earlier remove of Wilms tumor in a patient with heart failure must be considered. Only such a procedure - early surgical removal of the tumor on one side result in the improvement of cardiovascular function and the possibility of further chemotherapy.
  5. We add a paragraph about the genetics basis of Wilms tumor: Wilms tumor is genetically heterogeneous and mutations in WT1, WTX,and CTNNB1underlie the genetic basis of about one-third of Wilms tumors. Another gene mutation found in WIlms tumor is AMER1 Changes on the short  arm of chromosome 11 where IGF2 and H19 genes are located were also associated with developing of Wilms tumor.
  6. In the closest family to the second generation there was no cancer, including Wilms tumor. Nevertheless, an NGS examination was performed in the child with the evaluation of over 140 genes whose mutations predispose to cancer. The test was negative.
  7. ECHO examination was and is performed regularly in this child. EF% values at the time of diagnosis and further in the course of preoperative chemotherapy were 38-40%. After the first surgery and further in the course of postoperative chemotherapy, EF% increased to 50-55%. The last EF% after the completion of the entire postoperative chemotherapy was 60-65%. A paragraph on EF% changes has been added to the paper.
  8. The patient presented BP between 110/70 and 130/70 at the time of diagnosis. The child was receiving captopril, and spirinolactone.

Regards,

Danuta Januszkiewicz-Lewandowska, prof. MD

Patrycja Sosnowska, MD

Reviewer 3 Report

The manuscript "Bilateral nephroblastoma with dilated cardiomyopathy as an indication for off-protocol treatment: A case report" describes the case of a 9-month old girl, presenting with bilateral nephroblastoma, and a dilated cardiomyopathy. 

The interest of this case is clear: weighing the available evidence to make a decision for a staged surgery, limited chemotherapy and not for doxorubicin therapy. Also the mechanism for this dilated cardiomyopathy is clearly presented and justified by other reports on renal artery compromises and in this particular case also with plasma renin measurement (PRA), supporting the idea that the cardiomyopathy may have been the temporary result of angiotensin II intoxication.

Some minor comments

Textual

  1. Abstract (line 18) This publication aimed to present: this publication presents
  2. Abstract (line 20-21) The severe condition of the child forced the application of off-protocol treatment, i.e. accelerated resection of a larger tumor, enabled the improvement of heart performance and subsequent therapy: The severe condition of the child forced the application of off-protocol treatment, i.e. accelerated resection of a larger tumor, which enabled the improvement of heart performance and made subsequent therapy possible.
  3. Abstract (line 25) negatively affected the cardiovascular system: compromised the cardiovascular system in a young child.
  4. Introduction line 43: This publication aimed to present: This publication presents
  5. Introduciton line 47: A child is from.. : She was born from a second pregnancy at 39 weeks of gestation,
  6. Case report Line 52: CT of the abdomen proved: showed or demonstrated (Proof of tumor comes from histology).
  7. Case report Line 80: Uneventful nephrectomy followed by …. was performed: Uneventful nephrectomy was performed, followed by ..:
  8. Case report line 142-143: Regarding the order of intervention, the general rule suggested operating first, the more accessible side: Regarding the order of intervention, the general rule suggested that the more accessible side would be the one to operate on first.
  9. Case report line 143-144: In the described case, severe cardiomyopathy was a result of the larger mass and produced direct vital risk: In the described case, severe cardiomyopathy was a result of the larger mass on the right, less accessible side and produced a direct vital risk.  
  10. Case report line 145: That is why the life-saving resection of the bigger tumor in the presented case appeared proper: That is why the life-saving resection of the bigger tumor in the presented case appeared appropriate/ correct.      

Clarification

  1. Line 52: how high was the blood pressure, line 70: how high was the blood pressure.
  2. Line 71 and graph: why use the LDH? I expected a kidney clearance problem at the first stage, which then improved, was that the case ? Can this be inserted instead of LDH?
  3. Case report line 64: Coronary artery abnormalities were excluded: How were they excluded, by CT scan?
  4. Case report line 68: describe doses of vincristine and acinomycin
  5. Case report line 63 and 64: the FS is 10-12% and the LVEF is 38-40%. When I use Teicholz methods, I would calculate a LVEF of 26% from a FS of 12%. Was the LVEF measured by Biplane EF?
  6. Case report line 87-88: etoposide and carboplatin doses

Author Response

Dear Reviewer,

We apologize for the delay in responding to the Reviewer's comments. The only explanation for this is the multitude of duties in our work in the hospital and with medical students.

Thank you for your time to prepare a review of our article and for your insightful but valuable comments, especially on cardiomyopathy pathomechanism.

  1. All comments proposed by the Reviewer in points 1-10 have been corrected. In addition, the manuscript has been sent for linguistic correction, and we attach the manuscript with the corrections made and with version already corrected.
  2. At the time of diagnosis the patient presented BP between 110/70 and 130/70. The child was receiving captopril, and spirinolactone.
  3. We showed LDH values due to myocardial insufficiency features.
  4. Coronary artery abnormalities were excluded in Doppler ECHO and not in CT.
  5. LVEF values were given by the cardiologist performing the tests. LVEF was measured by biplane EF.
  6. We supplemented the doses of cytostatic drugs in the text (VCR 1.5mg/m^2, Actinomycin 0.045 mg/kg, Etoposide 150 mg/m^2 and Carboplatin 200 mg/m^2). All drug doses due to body weight below 12 kg according to SIOP scheme were reduced to 67%).

Regards,

Danuta Januszkiewicz-Lewandowska, prof. MD

Patrycja Sosnowska, MD

Round 2

Reviewer 1 Report

The manuscript revision was well done.

The authors have answered the questions about the data and revised the manuscript accordingly.

The revision is good and the manuscript is good for publication.

Reviewer 2 Report

Thank you for sharing an interesting case study with us and for replying to all of my comments. The manuscript deserves to be published after some minor revision, mostly pertaining to the English language and style. Wishing the authors all the best!